

# Psychometric properties of the Chinese version physical literacy assessment questionnaire among high school students in Gansu, China

Zilu Qu[1], Jiarun Wu[2], Yee Cheng Kueh[3], Dongqing Ye[1] and Garry Kuan[2]

[1] School of Physical Education, Guangzhou University, Guangzhou, Guangdong, China
[2] School of Health Sciences, Universiti Sains Malaysia, Kubang Kerian, Kelantan, Malaysia
[3] School of Medical Sciences, Universiti Sains Malaysia, Kubang Kerian, Kelantan, Malaysia

Corresponding authors
Dongqing Ye, ydqqzl@gzhu.edu.cn
Garry Kuan, garry@usm.my

## ABSTRACT

**Introduction.** The concept of physical literacy (PL) originated from the philosophy of lifelong sports, and its development is crucial for achieving lifelong sports goals. The Portuguese Physical Literacy Assessment Questionnaire (PPLA-Q) is a tool designed to assess the physical literacy of high school students, demonstrating good reliability and validity. The aim of this study is to translate and adapt the PPLA-Q into Chinese (PPLA-Q-C) and validate its applicability among adolescents in Gansu, China through confirmatory factor analysis (CFA).

**Methods.** A total of 780 middle school students from Gansu Province, China, aged 15–18, participated. After screening, the final sample size was 729. The English PPLA-Q was translated into Chinese and validated through expert review and pre-testing. Data were collected in September 2022 *via* a self-reported survey, with an effective response rate of 93.5%. Data analysis was conducted using Mplus 8.3 with the robust maximum likelihood (MLR) estimator. Confirmatory factor analysis (CFA) assessed model fit using standardized root mean square residual (SRMR), root mean square error of approximation (RMSEA), Tucker-Lewis index (TLI), and comparative fit index (CFI) indices. Construct validity was evaluated through composite reliability (CR), average variance extracted (AVE), and factor correlations. The study followed ethical guidelines and received approval from Universiti Sains Malaysia's ethics committee.

**Results.** All observed items exhibited high factor loadings, confirming that the assumed model, consisting of 40 items grouped into four latent variables, was consistent with the original model. The CFA model demonstrated a good fit with the data, as indicated by fit indices: RMSEA = .024 (90% CI [.020–.027]), CFI = .978, TLI = .977, SRMR = .032.

**Conclusion.** The final measurement model comprised 40 items, all retained and considered acceptable for the sample. The study results suggest that the Chinese version of PPLA-Q (PPLA-Q-C) is effective and reliable for assessing the PL of high school students in Gansu Province, China. Education practitioners and policymakers can utilize the PPLA-Q-C framework in their future work to assess the PL of Chinese-speaking students.

## INTRODUCTION

Physical literacy (PL) is a lifelong concept based on sport and physical activity, supporting individuals' engagement in physical exercise. It encompasses factors such as motivation, confidence, physical ability, understanding, and knowledge of each individual's participation in sports (*Australia, 2019*; *Whitehead, 2001*; *Whitehead, 2007*). PL is constantly changing throughout a person's life, but studies have shown that developing good PL habits at school age is crucial to the formation of PL (*Ares et al., 2024*; *Pangrazi & Beighle, 2019*; *Telama, 2009*; *Telama et al., 2015*). Children who actively engage in physical activities from an early age are more likely to develop and maintain healthy exercise habits that may persist into adulthood. This can help alleviate the widespread and growing issue of physical inactivity we currently face (*Guthold et al., 2018*; *Guthold et al., 2020*). Studies have shown that physical education in schools is considered to be a very important part of the process of developing PL in school-age children (*Liu & Chen, 2021*; *Nancy & Jannine, 2015*).

Physical education (PE) is a very special environment, as it has a certain mandatory aspect, requiring every ordinary student to participate and achieve the corresponding physical and technical standards (*Lieberman, Houston-Wilson & Grenier, 2024*). Additionally, there is no extra charge for the whole process, ensuring equal rights for everyone to participate (*Lieberman, Houston-Wilson & Grenier, 2024*). Moreover, professional PE teachers provide guidance and assistance throughout the process, ensuring proper development (*Admiraal et al., 2021*). Lastly, the equipment and venues used adhere to minimum safety standards, further enhancing the overall experience. Most importantly, due to various reasons, including limited resources, family environment, community conditions, and cultural factors, school physical education may represent the sole opportunity for some individuals to learn and engage in sports activities. Schools play a vital role not only in providing education, but also in fostering sports participation and PL. Therefore, for those unable to access other sports resources or opportunities, school physical education may serve as their only avenue for sports activities throughout their lives (*Onofre, 2017*; *Woods, Moyna & Quinlan, 2010*). Therefore, many studies have elaborated the importance and necessity of cultivating PL in school physical education classes (*Dudley, 2015*; *Dudley et al., 2017*). *Flemons, Diffey & Cunliffe (2018)* provided insights into the concept of PL and the challenges faced in fostering it within school physical education programs. He delineated definitions and practical approaches to promoting PL, suggesting measures to enable holistic development across various realms of physical activity. *Durden-Myers (2020)* underscored the pivotal role of PL in encouraging adolescents' participation in physical endeavors. The author emphasized that by cultivating PL, young individuals can acquire comprehension and proficiency in diverse sports activities, thereby boosting their motivation and eagerness to engage in physical pursuits. Furthermore, *Shimon (2019)* delved into essential considerations for devising policies concerning PL across sectors such as public health, recreation, sports, and education. They presented a range of policy recommendations aimed at fostering the integration and dissemination of PL across different spheres, ultimately contributing to improved overall health and well-being.

Numerous studies have shown that the development of PL can promote the achievement of lifelong physical education goals (*Whitehead, 2019*). However, certain questions remain unanswered regarding the development of PL, including the aspects from which it should be cultivated and the methodologies involved. Additionally, there is divergence among researchers regarding the sources of the components of PL (*Edwards et al., 2017*; *Liu & Chen, 2021*; *Martins et al., 2021*). For this reason, Australian scholars developed the Australian Physical Literacy Framework (APLF), providing a comprehensive model to understand PL holistically. It aims to assess and foster individuals' development in PL and guide both practice and policy. This framework comprises 30 different elements, categorizing PL into four domains: physical, cognitive, psychological, and social (*Australia, 2019*).

Studies have found that assessment plays a very important role in the development of PL in the process of implementation, so researchers have developed some assessment tools around how to assess PL (*Edwards et al., 2017*; *Liu & Chen, 2021*; *Shearer et al., 2021*). One of the most frequently used in other studies is Canadian Assessment of Physical Literacy (CAPL) (*Francis et al., 2021*; *Gunnell et al., 2018*), and the Physical Literacy Assessment for Youth (PLAY) (*Dudley et al., 2017*). CAPL is suitable for children aged 8–12 years, The questionnaire includes daily behaviour, physical competence, motivation and confidence, and knowledge and understanding (*Blanchard et al., 2020*). The PLAY scale is mainly designed for children aged 7 to 12, and its content includes three aspects: measures of motor competence, comprehension, and confidence. In actual use, both scales show high reliability and validity, but the tested sample is required to be children, younger than 12 years old, so the scope of application is limited (*Shearer et al., 2021*).

The Portuguese Physical Literacy Assessment Questionnaire (PPLA-Q) is a tool for assessing PL in high school students developed by Portuguese researchers in 2021 (*Mota, Martins & Onofre, 2021*). The whole scale includes 40 items. According to the relationship between the items, the developer divides the 40 items into four factors, namely motivation, confidence, emotional regulation and physical regulation. The PPLA-Q scale, with its sample age range set between 15–18 years, offers a more comprehensive approach compared to CAPL and PLAY. Notably, it compensates for the limitations of previous tools by incorporating aspects of psychological and social dimensions alongside physical and cognitive domains. Moreover, PPLA-Q stands out for its simplicity and ease of use, making it more accessible for practitioners. Furthermore, it demonstrates higher levels of reliability and validity (Cronbach's alpha = 0.87; CFI = 0.96; SRMR = 0.04) during its application (*Mota, Martins & Onofre, 2021*).

Based on the accuracy of measuring adolescent PL using the PPLA-Q, and the importance of fostering adolescent PL, there is a need for in-depth research and exploration. The purpose of this study was to translate the PPLA-Q into Chinese for use by Chinese high school students (aged 15–18 years, aligning with the standard age range of PPLA-Q), and then examine the reliability and validity of the Chinese version of the PPLA-Q for use by Chinese high school students. Currently, commonly used PL assessment tools in China include the Canadian Assessment of Physical Literacy - China (CAPL-C), Physical Literacy Assessment for Youth (PLAY), and Motor Skill Competence (MSC) (*Dlugonski*

*et al., 2022*; *Liu & Chen, 2021*). However, these tools have certain limitations. In terms of multidimensional assessment, CAPL-C, PLAY, and MSC each focus on different aspects of PL: CAPL-C emphasizes physical activity knowledge and attitudes, PLAY targets movement skills and participation behaviors, and MSC is dedicated to assessing fundamental motor skills (*Barnett et al., 2023*; *Delisle Nyström et al., 2018*; *Gao et al., 2019*). A single tool is limited in its ability to comprehensively cover the multidimensional concept of PL. In contrast, the PPLA-Q encompasses multiple factors—motivation, confidence, emotional regulation, and physical regulation—providing a more holistic evaluation perspective that is particularly beneficial for promoting the overall development of PL. This comprehensive assessment is crucial for enhancing the targeted effectiveness of PL education, an area where current tools are relatively limited. In terms of cultural adaptability, apart from CAPL-C, both PLAY and MSC are primarily designed based on Western adolescents' activity patterns, which may not fully align with the motivations and attitudes of Chinese youth. Furthermore, CAPL-C is primarily suitable for children aged 8–12 and does not effectively cater to Chinese adolescents (*Blanchard et al., 2020*). The PPLA-Q, on the other hand, is designed for adolescents aged 15-18, and its adaptation to Chinese cultural contexts considers the specific perceptions and behavioral characteristics of Chinese youth in relation to physical activity, providing a more accurate reflection of their PL levels. In terms of convenience, compared to the skill testing required by PLAY and MSC, the PPLA-Q utilizes a self-reported questionnaire, making it simpler to administer, and more cost-effective, thus suitable for large-scale application (*Cools, Samaey & Andries, 2009*; *Roach & Keats, 2018*). This feature is particularly advantageous in China, where the large population makes skill testing impractical (*Wang et al., 2016*). As a self-reporting tool, the PPLA-Q-C enables widespread PL assessments across schools and communities without additional labor costs. Overall, validating the PPLA-Q-C addresses gaps in multidimensional assessment, cultural adaptability, and broad applicability found in existing tools. It provides a practical, scientifically sound measure that supports the comprehensive evaluation and enhancement of PL among Chinese adolescents, offering schools and relevant educational departments a convenient tool for promoting PL development and fostering adolescents' physical and mental well-being. This underscores the importance and necessity of validating the PPLA-Q-C.

## METHOD

### Participants

All eligible students were invited to participate, resulting in a total of 780 participants in the study, and all of them came from five high schools in Gansu. After screening the submitted questionnaires, 729 students met the inclusion criteria. These criteria included being aged between 15 and 18 years old, having an average age of 16.2 years (SD = 1.1), being of Chinese nationality, having a strong understanding of Chinese with no difficulties in listening, speaking, reading, and writing, and being physically capable of participating in sports activities. Students with any physical ailments preventing their participation in sports were excluded. Of the final sample, 67.63% were males ($n = 493$) and 32.37% were females ($n = 236$).

The participants were selected using a random sampling method with voluntary participation, ensuring a diverse representation of students from various backgrounds and regions within Gansu Province. Situated at the geometric center of China, Gansu Province serves as a sample with a certain degree of representativeness due to its central location.

In general, in the process of confirmatory factor analysis (CFA), selecting a larger sample can make the validation results more stable and have higher reproducibility. Studies have shown that when the number of validations being processed is greater than six factors, it is best to require a sample size greater than 500 (*Hair et al., 2010*). In contrast to the current study, only four factors were included in the PPLA-Q-C questionnaire, so the sample size of 729 chosen for this study is completely sufficient for a reliable CFA.

### Questionnaire translation

The English version of PPLA-Q was translated into Chinese language using the following steps: (1) initial translation: the second and fourth authors translated the English version of PPLA-Q into Chinese according to the original meaning of the questionnaire. They were all experts proficient in both English and Chinese, possessing extensive experience in translating questionnaires. Both authors were sport psychologists with over 10 years of experience. They are of Chinese descent, fluent in both Chinese and English, and have a deep understanding of Chinese culture. (2) Back translation: translation by two additional Chinese individuals (one is a professor in English literature, and one is an associate professor in physical education) proficient in English, who back-translated the Chinese version into English. (3) Expert review: the english-to-Chinese and Chinese-to-English translations were evaluated by a panel of five experts (physical educator, two sport scientists, sport psychologist, and sport medicine doctor) with years of research in the relevant fields and extensive experience. They compared each translated item to its counterpart in the original English version, eliminating any possible differences in meaning. At the same time, the panel also assessed the cultural appropriateness of the translated content for the Chinese population. (4) Cultural appropriateness evaluation: the expert panel evaluated whether the content was culturally suitable for the Chinese population. (5) Pre-testing: the final Chinese version of the questionnaire (PPLA-Q-C) was pre-tested with a group of 10 students (the inclusion criteria remain consistent with the previous requirements.). These students answered the questions and provided feedback on the wording and presentation. The pre-test results indicated clarity and comprehension, with no modifications required.

### Data collection

Data collection was conducted in September 2022 at secondary schools in Gansu, China. This study utilized a cross-sectional design with self-administered PPLA-Q-C questionnaires. The process involved distributing questionnaires to students by class, with two researchers assigned to each class. Before distributing the questionnaires, researchers provided a brief introduction to the study's purpose and content. After signing informed consent forms, all participants voluntarily completed the questionnaires and returned them on the spot. Researchers checked the responses for errors and omissions to minimize

invalid questionnaires. Throughout the process, participants could consult researchers for clarification or assistance. A total of 780 PPLA-Q-C questionnaires were distributed, with a 100% response rate. Due to careful checks during collection, there were 51 invalid questionnaires, resulting in an effective response rate of 93.5%. All participants' information is strictly for the use of this study only. It is strictly prohibited to disclose information to non-participants, and all collected data will be handed over to a dedicated person in the research team for safekeeping after collection.

## MEASURES

### Demographic

This study included some simple demographic information, and the questions included age, gender, grade and class in school. In future studies, these data can be used to further compare differences in PL among participants of different ages, genders, and grades.

### Portuguese Physical Literacy Assessment Questionnaire

The Portuguese Physical Literacy Assessment Questionnaire (PPLA-Q) is a survey designed and developed by *Mota, Martins & Onofre (2021)*. The function of this questionnaire is to measure the PL of high school students aged 15–18. The questionnaire contained 40 items and four factors (motivation, confidence, emotional regulation, and physical regulation). Scoring is done on a 5-point Likert scale from 0 (not at all) to 4 (totally). In this study, researchers utilized a Chinese-translated version of the PPLA-Q to validate into the Chinese version (PPLA-Q-C) (method was stated in above).

### Ethics approval

The study was conducted in accordance with the Declaration of Helsinki (*World Medical Association, 2013*), and was approved by the Human Research Ethics Committee of Universiti Sains Malaysia (USM/JEPeM/22040247). Prior to the study, participants provided written informed consent. All participants completed the PPLA-Q-C questionnaire after voluntarily completing the written consent form and returned it to the researcher. The researchers ensured that participants kept their answers to the questionnaire confidential and were informed of their right to withdraw from completing the questionnaire at any stage without penalty.

### Statistical analysis

The data analysis is conducted using Mplus 8.3. If the collected data contain missing values, a missing data model should be employed for analysis. If there are no missing values in the collected questionnaires, then all the data can be used for subsequent analysis. In confirmatory factor analysis (CFA), if the data adhere to the assumption of normality (verified through Mardia's multivariate skewness and kurtosis tests with $p$-values exceeding 0.05), the maximum likelihood (ML) estimator should be used for model fitting. Additionally, unbiased standardized estimates of path coefficients should be provided. If the data deviate from the normality assumption, the robust maximum likelihood (MLR) estimator should be utilized for model fitting. This enhances the robustness of the model

**Table 1  Summary of the goodness of fit indices cut-off values.**

| Fit indices | Symbol | Cut-off value | Suggested by | Comments |
|---|---|---|---|---|
| Comparative fit index | CFI | >0.95, good fit<br>between 0.90 to 0.95, reasonable fit | *Hair Jr et al. (2014)* | For sample size more than 250;<br>a) If m < 12, CFI/TLI ≥ 0.95 |
| Tucker Lewis index | TLI | >0.95 | | b) If 12 < m < 30, CFI/TLI > 0.92 |
| Root mean square of approximation | RMSEA | <0.05, good fit<br><0.08, reasonably fit<br><0.10, poor fit | *Hair Jr et al. (2014)* | |
| Standardised root mean square residual | SRMR | <0.08 | | |

**Notes.**

m, number of observed variables.

against deviations from normality in the data distribution, providing relatively stable parameter estimates.

The initial proposed measurement model was developed and evaluated through CFA. This model encompassed four latent variables corresponding to the subscales of the PPLA-Q-C, along with 40 observed variables representing items within the questionnaire. To guide the retention or removal of items from the measurement model, factor loadings of 0.40 and above were considered, along with significance in *p*-values and modification indices. This approach follows the recommendations of *Ford, MacCallum & Tait (1986)* and *Wang & Wang (2019)*.

This study conducted a thorough assessment of the validation results, employing a set of multiple fit indices that needed to meet specific criteria to determine the adequacy of the model fit. Generally, comparing various fit indices, including a comparative fit index (CFI) and Tucker and Lewis index (TLI) exceeding 0.95, root mean square error of approximation (RMSEA) lower than 0.07 and standardized root mean square residual (SRMR) lower than 0.08, simultaneously meeting the recommended thresholds suggests a favorable model fit. In such cases, the results can be retained (*Hair et al., 2010*; *Kline, 2023*).

The fit indices used in CFA were presented in Table 1.

Subsequent to identifying the best-fit measurement model, the construct validity of the four factors was assessed. Construct validity within CFA encompasses two facets: convergent and discriminant validity. Convergent validity examines the extent to which items within the same factor share a substantial portion of common variance. This assessment utilized composite reliability (CR) and average variance extracted (AVE). CR, computed using Raykov's method, gauges the scale's reliability (*Raykov & Marcoulides, 2016*). CR is recommended over Cronbach's alpha for reliability assessment in CFA models (*Raykov, 2001*). A CR value of 0.60 or higher is generally acceptable, along with an AVE value of 0.50 or higher (*Fornell & Larcker, 1981*; *Tseng, Dörnyei & Schmitt, 2006*).

Discriminant validity, which examines the distinctiveness of each factor from the others, was evaluated by inspecting correlations between factors in the model. According to *Brown (2015)*, discriminant validity can be established if correlation coefficients between factors are not excessively high (≤ 0.85).

**Table 2  Goodness of fit indices for physical literacy with four factors (standard values and actual values).**

| Model fitting index | $\chi^2$ | df | $\chi^2$/df | SRMR | RMSEA | TLI | CFI |
|---|---|---|---|---|---|---|---|
| Cut-off value | | | <5 | <0.08 | <0.08 | >0.95 | >0.95 |
| Actual value | 1,031.035 | 734 | 1.405 | 0.032 | 0.024 | 0.977 | 0.978 |

Notes.

$\chi^2$, Chi-Square; df, Degrees of Freedom; CFI, Comparative Fit Index; TLI, Tucker-Lewis Index; SRMR, Standardised Root Mean Square Residual; RMSEA, Root Mean Square Error of Approximation.

# RESULTS

## Measurement model PPLA-Q-C

Under the premise of not altering the original intent of the questionnaire, strictly following the previously mentioned method for questionnaire translation, the PPLA-Q were translated into PPLA-Q-C. Due to the absence of any missing values in the collected questionnaires, all gathered data can be utilized for subsequent analysis. Additionally, the current study conforms to the assumption of normality (confirmed through Mardia's multivariate skewness and kurtosis tests, with $p$-values exceeding 0.05). Therefore, the maximum likelihood (ML) estimator has been chosen for the subsequent confirmatory factor analysis (CFA).

The outcomes of all the examined models are outlined in Tables 2 and 3. The envisaged measurement model for PPLA-Q-C comprises four factors, encompassing a total of 40 items, distributed evenly as 10 items per factor. The factor loadings for all the items are detailed in Table 3.

It can be seen from the data in Tables 2 and 3 that the model fit of PPLA-Q-C is good, with factor loadings of all projects exceeding 0.6.

## Convergent and discriminant validity

To test the construct reliability of hypothesized models, we calculated CR values. The calculated CR values ranged from 0.908 to 0.922, which indicated the good reliability of the structure. Also assume that the AVE of each factor in the model ranges from 0.498 to 0.542. While certain factors displayed AVE values beneath the recommended threshold of 0.50, the CR values surpassed the suggested threshold of 0.60. This enables us to assert that the construct's convergent validity is acceptable (*Fornell & Larcker, 1981*). Furthermore, all inter-factor correlations remained below the advised limit of 0.85, affirming strong discriminant validity. The corresponding CR, AVE, and correlation coefficient values are detailed in Table 4.

# DISCUSSION

The formulation of PPLA-Q-C is an important step in order to gain a deeper understanding of the Chinese-speaking population's participation in PL. This study aimed to perform a confirmatory test on the factor structure of PPLA-Q-C. After performing confirmatory factor analysis on PPLA-Q-C, we found that it showed good performance in terms of both factor loadings and model fitting. Therefore, we retained all the four factors and
**Table 3  Standardised factor loadings for model.**

| Factors and items | Factor loadings |
| --- | --- |
| Motivation | |
| Q1. I am motivated to practice PA | 0.735 |
| Q2. I practice PA because others tell me I should | 0.774 |
| Q3. I feel guilty when I do not practice PA | 0.708 |
| Q4. I feel bad about myself when I do not practice PA | 0.763 |
| Q5. I feel pressured by others to practice PA | 0.672 |
| Q6. I practice PA because I feel others would be unhappy if I did not | 0.633 |
| Q7. I practice PA because it is fun | 0.698 |
| Q8. I feel good when I practice PA | 0.736 |
| Q9. I consider PA a part of me | 0.694 |
| Q37. I recognize my physical limits | 0.712 |
| Confidence | |
| Q10. I value the benefits of PA | 0.744 |
| Q11. I see PA as a fundamental part of who I am | 0.814 |
| Q12. I enjoy practicing PA | 0.696 |
| Q13. I feel confident to practice PA | 0.774 |
| Q14. I am confident in my abilities | 0.749 |
| Q15. I can participate with success | 0.776 |
| Q16. I consider myself competent | 0.698 |
| Q17. I trust my skills | 0.621 |
| Q18. I feel good about the way I am able to participate | 0.680 |
| Q38. I can recognize the effect that different intensities have in me | 0.787 |
| Emotional Regulation | |
| Q19. I can participate in PA that I consider challenging | 0.646 |
| Q20. I know how to become more confident in myself | 0.751 |
| Q21. I feel competent even when I am criticized | 0.660 |
| Q22. I believe in myself even when I lose | 0.701 |
| Q23. I can manage my emotions | 0.716 |
| Q24. I can recognize other's emotions | 0.736 |
| Q25. I can recognize my emotions | 0.822 |
| Q26. I am sensitive to the feelings of others | 0.711 |
| Q27. I understand what others feel | 0.718 |
| Q39. I use strategies to manage my effort | 0.713 |
| Physical Regulation | |
| Q28. I can identify what I feel | 0.709 |
| Q29. I can anticipate what I will feel | 0.831 |
| Q30. I can deal with difficulties rationally | 0.744 |
| Q31. I can manage my emotions when necessary | 0.739 |
| Q32. I have a good control of my emotions | 0.631 |

| Factors and items | Factor loadings |
|---|---|
| Q33. I can manage my effort | 0.723 |
| Q34. I know when I am tired | 0.619 |
| Q35. I can recognize changes in my breathing | 0.714 |
| Q36. I can recognize changes in my heart rate | 0.616 |
| Q40. I recognize my physical limits | 0.702 |

**Table 4  PPLA-Q-C reliability and discriminate validity.**

| Dimension (DIM) | Composite reliability | Convergence validity | Discriminate validity | | | |
|---|---|---|---|---|---|---|
| | CR | AVE | MOT | CON | EMO | PHY |
| MOT | 0.912 | 0.509 | **0.713** | | | |
| CON | 0.922 | 0.542 | 0.480[*] | **0.736** | | |
| EMO | 0.914 | 0.517 | 0.526[*] | 0.456[*] | **0.719** | |
| PHY | 0.908 | 0.498 | 0.536[*] | 0.484[*] | 0.548[*] | **0.706** |

Notes.

The diagonal bold is the AVE square root value, and the lower triangle is the pearson correlation of factors.

*Correlation is significant at the 0.01 level (two tailed).

40 questions in the original questionnaire, and no questions needed to be deleted. It is worth noting that previous studies have shown that the original version of PPLA-Q has reliability, validity and stability over time. The four core factors of the PPLA-Q-C— motivation, confidence, emotional regulation, and physical regulation—are closely aligned with the concept of physical literacy. Firstly, motivation serves as the driving force behind participation in physical activities, directly influencing adolescents' sustained interest and engagement (*Ke & Jiang, 2024*). For adolescents aged 15–18, a period when self-awareness is increasingly prominent, cultivating motivation not only encourages active participation in physical activities but also enhances their acceptance of a healthy lifestyle (*Hajar et al., 2019*). Secondly, confidence represents the self-assurance needed to face challenges and overcome difficulties in sports (*Feltz, Short & Sullivan, 2008*; *Markway & Ampel, 2018*). Adolescents in this age group are experiencing rapid physical development, and building confidence supports their ability to master and refine movement skills, thereby fostering comprehensive physical literacy (*Chen, 2022*). Emotional regulation plays a crucial role in the mental health of adolescents, especially given the heightened emotional variability typical of this age (*Wu et al., 2024*). Physical activity provides an outlet for emotional release, promoting healthier emotional management (*Martín-Rodríguez et al., 2024*). Lastly, physical regulation, the ability to control physical activity, is central to physical literacy's core goal (*Durden-Myers, Green & Whitehead, 2018*). Adolescents aged 15–18 are at a pivotal stage of physical development, and strengthening their physical regulation supports their ability to perform complex movements with greater ease. The PPLA-Q-C's measurement of these four factors offers a comprehensive approach to capturing the characteristics of physical literacy, providing educators with key data to understand and support the development of physical literacy in adolescents. In addition, PPLA-Q is also

related to the key of self-determination theory (SDT) (*Mota, Martins & Onofre, 2021*). This includes motivation research and support environment. Motivation research suggests that PPLA-Q-C can provide data relevant to SDT, helping researchers understand the motivations of Chinese students to participate in physical activities, further exploring the relationship between intrinsic motivation, autonomy, and participation in sports (*Howard et al., 2018*). In addition, in terms of support environment, the use of PPLA-Q-C can assess whether the environment Chinese students are in supports the development of their PL, thereby providing important information for researchers and educators to improve the environment for the development of students' PL (*Luo et al., 2022*). In the context of Chinese culture, the application of PPLA-Q-C takes into account specific cultural values and social environments, which are crucial for a deeper understanding of the development of PL among Chinese students and the applicability of SDT in China (*Carl et al., 2023*). Therefore, further research on how to strengthen or expand SDT within the Chinese cultural context to better explain the development of PL among Chinese students would be beneficial. Therefore, we translated the original English PPLA-Q into the Chinese version PPLA-Q (PPLA-Q-C) to better meet the needs of local residents, because Chinese is the most commonly used and easy-to-understand language in China.

Since the participants in this study were of the same age as those in the original PPLA-Q, and PPLA-Q-C showed high model fit and reliability and validity, we can conclude that PPLA-Q-C can be well applied to Chinese students. The method used in this study to test and confirm the factor structure of PPLA-Q-C is confirmatory factor analysis. This method was chosen because it enables the evaluation of the validity of measurement models constructed from measurement theory (*Mota et al., 2023*). Assuming that the structure of PPLA-Q-C is the same as PPLA-Q, it consists of four factors, all with 10 items. Once the factor structure of the PPLA-Q-C has been confirmed, it becomes possible to evaluate its construct validity. Construct validity pertains to the extent to which a set of measurement items genuinely represent the theoretical factors within the construct (*Bagozzi et al., 1991*). In simpler terms, it gauges the precision of the measurement (*Hair et al., 2010*).

It was previously stated in the Results section that two aspects of the construct validity of the PPLA-Q-C were assessed, namely convergent validity and discriminant validity. The paragraph elaborates on the findings regarding these aspects. The average variance extraction (AVE) was computed, reflecting the average of the squared factor loadings for each factor, with values ranging from 0.498 to 0.542 in this study. Despite some AVE values being slightly below the conventional threshold of 0.50, it is noted that 0.498 is very close to 0.50. Moreover, the convergence validity of the structure remains acceptable as long as the composite reliability (CR) exceeds 0.60 (*Fornell & Larcker, 1981*; *Huang et al., 2013*). The paragraph further highlights that the construction reliability based on CR ranges from 0.908 to 0.922 in this study, surpassing the recommended threshold of 0.60 proposed by *Tseng, Dörnyei & Schmitt (2006)*. Additionally, it emphasizes the robust support for discriminant validity, as evidenced by the correlation coefficients among the factors remaining below the suggested threshold of 0.85. This supports the assertion that each factor in the PPLA-Q-C is distinct, without significant overlap with other factors, and

captures unique phenomena. This reflexive explanation contextualizes the significance of the reported findings within the broader methodological framework of the study.

## Research limitation

However, we must also admit that there are some limitations in this study. First, all the data were collected from Lanzhou, Gansu Province, so it is not known whether PPLA-Q-C is suitable for high school students in other parts of China, which may limit the generalization of the findings to students in other parts of China. A recommendation for future research would be to conduct cross-regional validation studies to assess the applicability and effectiveness of PPLA-Q-C among high school students across various regions of China. This would enhance the robustness and validity of the assessment tool, facilitating its broader implementation and ensuring its relevance for a diverse population of Chinese students. The second limitation is that all survey results are based on participants' self-reports, without other forms of corroboration, and self-reports may have a high degree of autonomy, which may reduce the accuracy of the complete data. In the end, participants may be influenced by a reflection of social expectations, and they may fill out answers that make them look better (*He & Van de Vijver, 2013*). Therefore, we can only ask participants to fill in as truthfully as possible. Furthermore, this study did not delve further into utilizing additional sociodemographic data. These aspects will be addressed in subsequent research endeavors.

In this study, the PPLA-Q-C is the same as the PPLA-Q, also confirmed the four-factor structure, and showed good construct validity. Future research should focus on the replicability of PPLA-Q-C in other regions where Chinese is the main language. Researchers ought to further explore the temporal dynamics and consistency of both PPLA-Q and PPLA-Q-C through longitudinal investigations. Longitudinal measurements present several benefits, including the capacity to yield richer insights compared to cross-sectional studies (*Lynn, 2009*). This enables researchers to scrutinize evolving patterns and variability, assess the extent of measurement invariance, and explore potential causal connections (*Koch et al., 2014*).

## Research prospect

The research implications of validating the PPLA-Q-C are significant. Research result indicates that the PPLA-Q-C is confirmed to be a valid tool for measuring physical literacy among Chinese-speaking adolescents, it provides researchers and educators with a culturally adapted, reliable instrument to evaluate core components of physical literacy. This enables longitudinal tracking of adolescents' development in motivation, confidence, emotional regulation, and physical regulation, thus offering deeper insights into their physical literacy journey. Validating this tool also allows policymakers and educators to tailor interventions that address specific needs in physical education programs, aiming to enhance students' physical literacy comprehensively. Furthermore, a valid instrument can facilitate comparative studies across different cultural contexts, contributing to global physical literacy research and establishing benchmarks for physical literacy in China.

In future educational settings, researchers can utilize PPLA-Q-C to assess and cultivate the PL of Chinese-speaking students. This involves understanding their levels of PL

across the domains of physical, cognitive, psychological, and social aspects. Based on assessment results, individual strengths and needs of students can be identified, leading to the development of personalized development plans. Additionally, appropriate support and resources such as extracurricular sports activities, sports training courses, and individual coaching can be provided to aid students in enhancing their PL. Simultaneously, researchers can gain insights into students' motivations and levels of autonomy in sports participation through assessment results. By encouraging voluntary participation in sports activities and emphasizing the importance of physical exercise for overall health and well-being, students can be motivated to engage in sports. Finally, researchers can conduct regular monitoring and assessment of students using PPLA-Q-C to track their development in PL and adjust educational strategies and support measures as needed. Through these approaches, effective utilization of PPLA-Q-C in assessing and fostering the PL of Chinese-speaking students can promote their holistic development and enhance their physical and mental well-being.

## CONCLUSION

In this study, by verifying that the final measurement model of the PPLA-Q-C questionnaire is the same as the null hypothesis, all items are suitable for the sample of this study, and all items are retained. In future research, if it is necessary to measure the PL of high school students, we can consider using PPLA-Q-C to explain their state within the framework of four factors in the population whose main language is Chinese.

## ACKNOWLEDGEMENTS

We would like to express our sincerest gratitude to all the participants who contributed to the study.

### Funding

This research was supported by the Ministry of Higher Education Malaysia for the Fundamental Research Grant Scheme (FRGS) with project code: FRGS/1/2020/SKK06/USM/03/13. The funders had no role in study design, data collection and analysis, decision to publish, or preparation of the manuscript.

### Grant Disclosures

The following grant information was disclosed by the authors:
The Ministry of Higher Education Malaysia for the Fundamental Research Grant Scheme (FRGS): FRGS/1/2020/SKK06/USM/03/13.

### Competing Interests

ZQ, JW, GK, DY and KYC declared no conflict of interest.

## Author Contributions

- Zilu Qu conceived and designed the experiments, prepared figures and/or tables, authored or reviewed drafts of the article, and approved the final draft.
- Jiarun Wu conceived and designed the experiments, performed the experiments, analyzed the data, prepared figures and/or tables, authored or reviewed drafts of the article, and approved the final draft.
- Yee Cheng Kueh analyzed the data, authored or reviewed drafts of the article, and approved the final draft.
- Dongqing Ye conceived and designed the experiments, prepared figures and/or tables, authored or reviewed drafts of the article, and approved the final draft.
- Garry Kuan conceived and designed the experiments, performed the experiments, prepared figures and/or tables, authored or reviewed drafts of the article, and approved the final draft.

## Human Ethics

The following information was supplied relating to ethical approvals (*i.e.,* approving body and any reference numbers):

The study was conducted in accordance with the Declaration of Helsinki (World Medical Association, 2013), and was approved by the Human Research Ethics Committee of Universiti Sains Malaysia (USM/JEPeM/22040247).

## Data Availability

The raw measurements are available in the Supplementary File.

## Supplemental Information

Supplemental information for this article can be found online at http://dx.doi.org/10.7717/peerj.19093#supplemental-information.

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
