# Peer review of "Psychometric properties of the Chinese version physical literacy assessment questionnaire among high school students in Gansu, China"

_PeerJ, doi:10.7717/peerj.19093_

## Round 0.1 · original submission · Major Revisions

The authors are requested to carefully revise the manuscript and answer the questions raised by the reviewers.

The concerns of Reviewer 1 require significant effort to address.

Reviewer 1 ·

Basic reporting

This study replicates the physical literacy assessment questionnaire to examine the psychometric properties of the Chinese student population. I understand the necessity of replication studies and examining psychometric characteristics in another geographical location. However, replication should also make some unique contributions. How should applying physical literacy assessment differ in China, and why? The findings from this study are nothing new.

Experimental design

My primary concern is that the research reported here doesn't have the potential to provide substantial new contributions to the discipline. Thus, I need help to see this manuscript's theoretical and practical impact. For any work to be of value to the readers, it needs to go beyond what has already been done and extend our knowledge significantly. While I understand much effort must have gone into the manuscript, I encourage you to delve into the topic to establish a solid conceptual framework and provide additional insights. This paper is a good base for further research examining the holistic approach to physical literacy in China. Since you found that the examined tool has good psychometric characteristics, use it to explain the situation in the Chinese population and compare it with other authors. In this way, you can ensure better visibility of the research results.

Validity of the findings

As the above sections demonstrate significant challenges, it is inappropriate to review this section and the others (discussion, research prospect, and conclusion). In general, I suggest that this manuscript should be rejected.

Reviewer 2 ·

Basic reporting

Please ensure that all abbreviations are written in full at the very beginning, starting from the abstract section. For example: TLI, RMSEA, and SRMR
Please recheck the citation of Whitehead (lines 69 & 545) to ensure whether they refer to the same author.
The term "Physical Literacy" is used excessively, and its capitalization is inconsistent (lines 69-71). Please revise it to ensure consistent usage, either in upper and lower case or all in lower case.
It would be better to remove the phrase "For example" (line 92) and directly present the statement or research findings.
Please recheck the citation "Australia, 2019" (lines 69, 115, 425) to determine whether it refers to an institution or the country.
There is a typo in the abbreviation CPLA (line 123).
The term "Physical literacy assessment" (line 117) should be integrated into the introduction instead of being written as a separate subheading.
Describe the real conditions or existing secondary data from the research location, such as how physical literacy assessments have been conducted so far, including any limitations, weaknesses, or other aspects. Therefore, it should not immediately conclude that such assessments do not exist and aim to adapt the physical literacy assessment from Portugal.

Experimental design

In the methods section, how many schools are represented by the 780 participants?

Validity of the findings

In Table 2, the column headings for the abbreviations SRMR, RMSEA, TLI, and CFI should remain consistent with the order presented in the abstract and the description below Table 2.
In Table 3 the abbreviation "PA" needs to be explained.
In the discussion, there should also be a focus on the four groups of variables in the questionnaire instrument "(Motivation, Confidence, Emotional Regulation, Physical Regulation)", and then elaborate on these with the concept of physical literacy and the characteristics of students aged 15-18.
Explain the research implications if the physical literacy questionnaire instrument is valid.

Additional comments

Overall, this manuscript is well-written, but it still needs a thorough check for punctuation, word usage, and standard sentence structures. The introduction should also elaborate on the real conditions or secondary data from the research location, explaining why it is necessary to adopt the physical literacy assessment from Portugal. Avoid immediately stating that there is no specific tool available in China to assess physical literacy (line 146). Clarify how many schools the participants represent and explain the implications of this research.

---

## Round 0.2 · Minor Revisions

The authors are requested to carefully revise the manuscript and answer the questions raised by the reviewers.

Reviewer 2 ·

Basic reporting

Part of the manuscript has been revised according to the suggestions provided in response. Please feel free to proceed with publication if everything is satisfactory.

Experimental design

Part of the manuscript has been revised according to the suggestions provided in response. Please feel free to proceed with publication if everything is satisfactory.

Validity of the findings

Part of the manuscript has been revised according to the suggestions provided in response. Please feel free to proceed with publication if everything is satisfactory.

Reviewer 3 ·

Basic reporting

None.

Experimental design

1. What's sampling method of the current study?
2. Line 156, "729 students met the criteria", what is the criteria?

Validity of the findings

1. It is recommended to indicate the full name of the English abbreviation at the bottom of the table.
2. AVE value in Table 4 is a bit low.

Additional comments

None

---

## Round 0.3 · accepted · Accept

After revisions, one reviewer agreed to publish the manuscript. There is one reviewer left with a minor revision, and I think the author has responded adequately. I also reviewed the manuscript and found no obvious risks to publication. Therefore, I also approved the publication of this manuscript.